# Absence of an Association between Macular Degeneration and Young-Onset Dementia

**DOI:** 10.3390/jpm12020291

**Published:** 2022-02-16

**Authors:** Tung-Mei Tammy Kuang, Sudha Xirasagar, Wei-Yun Lee, Yen-Fu Cheng, Nai-Wen Kuo, Herng-Ching Lin

**Affiliations:** 1Department of Ophthalmology, Taipei Veterans General Hospital, Taipei 112, Taiwan; kuangtammy@gmail.com; 2Department of Ophthalmology, School of Medicine, National Yang Ming Chiao Tung University, Taipei 112, Taiwan; 3Research Center of Sleep Medicine, College of Medicine, Taipei Medical University, Taipei 110, Taiwan; 4Department of Health Services Policy and Management, Arnold School of Public Health, University of South Carolina, Columbia, SC 29208, USA; sxirasagar@sc.edu; 5School of Health Care Administration, College of Management, Taipei Medical University, Taipei 110, Taiwan; m911108007@tmu.edu.tw (W.-Y.L.); nwkuo@tmu.edu.tw (N.-W.K.); 6Department of Medical Research, Taipei Veterans General Hospital, Taipei 112, Taiwan; entist@gmail.com; 7Department of Otolaryngology-Head and Neck Surgery, Taipei Veterans General Hospital, Taipei 112, Taiwan; 8Department of Otolaryngology-Head and Neck Surgery, School of Medicine, National Yang Ming Chiao Tung University, Taipei 112, Taiwan; 9Sleep Research Center, Taipei Medical University Hospital, Taipei 110, Taiwan

**Keywords:** young onset dementia, macular degeneration, epidemiology

## Abstract

A few population-based studies have reported an association between prior age-related macular degeneration and senile dementia. No study has explored a possible link between prior macular degeneration and young-onset dementia (YOD). This case–control study aimed to evaluate the association of YOD with prior macular degeneration diagnosed in the 5-year period before their index date. Data for this retrospective observational study were retrieved from Taiwan’s National Health Insurance (NHI) dataset. A total of 36,577 patients with newly diagnosed YOD from January 2010 to December 2017 were identified as the study cohort, assigning their diagnosis date as their index date. Comparison patients were identified by propensity score-matching (three per case, *n* = 109,731 controls) from the remaining NHI beneficiaries of the period, their index date being the date of their first ambulatory care claim in the year of diagnosis of their matched YOD case. Chi-square test revealed no significant difference in the prevalence of prior macular degeneration between cases and controls (1.1% vs. 1.0%, *p* = 0.111). Conditional logistic regression analysis also showed an unadjusted odds ratio (OR) for prior macular degeneration of 1.098 among cases relative to controls (95% CI: 0.9797–1.232). Adjusted analysis confirmed that YOD was not associated with prior macular degeneration, adjusted odds ratio 1.098 (95% CI = 0.979–1.232). We conclude that patients with macular degeneration are not at increased risk for YOD.

## 1. Introduction

Dementia is an increasingly prevalent public health problem. Over 55 million people are estimated to have dementia worldwide, which is expected to increase to 78 million by 2030 [1]. The prevalence of dementia in the United States is 14.4% among the elderly aged over 68 years [2]. The dementia burden continues to grow due to an ageing society worldwide [3]. Dementia causes serious functional impairment at any age, and can be disastrous for patients and families when it sets in at a young age. Young-onset dementia (YOD) is usually defined as dementia with onset before the age of 65 years. The estimated prevalence rates of YOD in various studies are much lower, about 42–77 per 100,000 population in the 30–65 age-group, and 98–163.1 per 100,000 in the 45–64 group [4,5,6,7]. Senile dementia and YOD share many risk factors, such as age [8,9], sex [10,11], smoking [12,13,14], alcohol use [15,16,17], stroke [18,19], traumatic brain injury [14,20,21], cardiovascular diseases [22,23], diabetes mellitus [24,25], obesity [26,27], dyslipidemia [28,29], and hereditary factors [30,31].

Age-related Macular Degeneration (AMD), the third leading cause of severe vision loss, is a disease that involves the macular region of the retina, resulting in progressive loss of central vision. In 2016, the number of people with AMD globally was estimated at about 176 million, and this is expected to increase to nearly 288 million by 2040 [32]. Therefore, AMD is a major public health problem causing substantial socioeconomic burden.

A few empirical studies have attempted to investigate the associations between AMD and dementia. One study from the Korean National Health Insurance Service—Health Screening Cohort found that AMD patients had a higher risk for Alzheimer disease (AD) compared to non-AMD participants. They also observed that the association between AMD and AD was sustained even among those with healthy lifestyle behaviors [33]. Another large-scale, population-based study reported that AMD, especially nonexudative AMD, is independently associated with an increased risk of subsequent AD or senile dementia [34]. One study also observed that dementia was associated with a higher likelihood of prior neovascular AMD than comparison patients without dementia [35]. Finally, one systematic review of studies documented during the period of 1959–2018 captured from a wide range of indexing databases (MEDLINE, EMBASE, Web of Knowledge, PsycInfo and the Cochrane database) concluded that AMD was significantly associated with increased risk of AD/cognitive impairment [36].

In contrast, a population-based, cross-sectional study of 1179 participants aged 60 to 80 years from the Singapore Malay Eye study failed to detect a significant independent association between AMD and cognitive dysfunction [37]. Another study from the English National Health Service also found that the likelihood of receiving a diagnosis of AD or other dementia after having had a prior diagnosis of AMD was no different from that expected by chance [38]. In addition, one study using a random sample of the US Medicare population found no association of low scores on the Modified Mini-Mental State Examination, dementia, or AD with early AMD [39]. Therefore, the association between AMD and dementia remains unclear.

A few studies have explored some possible underlying pathogenic mechanisms shared between AMD and senile dementia. One study examining the sequence of a septet of sncRNAs (miRNA-7, miRNA-9-1, miRNA-23a/miRNA-27a, miRNA-34a, miRNA-125b-1, miRNA-146a and miRNA-155) and their abundance reported that these were significantly increased and abundant in both the AD-affected superior temporal lobe neocortex and the AMD-affected macular region of the retina [40]. Another study found extensive similarity in the prevailing immune and inflammatory degenerative mechanisms between AMD and AD [41]. One study suggested that AD and AMD share common features such as vitronectin and amyloid-β accumulation, increased oxidative stress, and the apolipoprotein and complement activation pathways [42]. In addition, one study summarized recent findings on the shared characteristics and perspectives between AMD and AD. They reported that an important characteristic common to both diseases is the presence of amyloid β (Aβ) in the senile plaques of the brain among patients with AD and in the drusen of AMD patients [43].

Some uncertainty continues about potential associations between AMD and dementia. Further, although several studies report significant associations between AMD and dementia, to our knowledge, no study has documented the relationship, if any, between macular degeneration and YOD to date. In particular, the noticeably increased incidence of YOD has drawn much attention during the past decade. Uncovering such an association, if any, may contribute to clinical guidelines for follow up care of younger patients with macular degeneration. The present study aimed to document YOD occurrence among a population-wide cohort of patients diagnosed with macular degeneration using a population-based, retrospective case–control study design.

## 2. Methods

### 2.1. Database

Data were retrieved from the Taiwan National Health Insurance (NHI) Research Database (NHIRD). Taiwan implemented its NHI program in 1995. Each year, the Bureau of NHI collects and curates claims data from the NHI program into the NHIRD, followed by de-identification of both patients and medical facilities, rendering it impervious to identification of subjects or institutions by researchers who are provided the data for research. The NHIRD consists of registration files and medical claims data of approximately 99% of Taiwan’s population (about 24.02 million as of December 2021) covered by the NHI program. The registration files include the registries of beneficiaries, contracted medical facilities, board-certified specialists, and medical personnel. The medical claims data include diagnoses, inpatient expenditures by admissions, details of inpatient orders, ambulatory care expenditures by visits, and details of ambulatory care orders. Many researchers in Taiwan have used data from the NHIRD for clinical-epidemiologic studies to identify associations between diseases and treatment outcomes. The NHIRD offers a unique opportunity to explore the association of young-onset dementia with macular degeneration.

The study was approved by the institutional review board of Taipei Medical University (TMU-JIRB N202009055), which complies with the Declaration of Helsinki. Informed consent was waived because the study used retrospective administrative claims data.

### 2.2. Identification of Cases and Controls

We retrieved 43,582 patients aged <65 years old with a first-time diagnosis of young-onset dementia during an ambulatory care visit between 1 January 2010 and 31 December 2017. We identified them using ICD-9-CM codes 290.0 (senile dementia, uncomplicated), 290.10 (presenile dementia, uncomplicated), 290.11 (presenile dementia with delirium), 290.12 (presenile dementia with delusional features), 290.13 (presenile dementia with depressive features), 290.20 (senile dementia with delusional features), 290.21 (senile dementia with depressive features), 290.3 (senile dementia with delirium), 290.4 (arteriosclerotic dementia), 294.1 (dementia in conditions classified elsewhere), 331.0 (Alzheimer’s disease), 331.1 (Pick’s disease), or ICD-10-CM codes F02.80 (dementia in other diseases classified elsewhere without behavioral disturbance), F02.81 (dementia in other diseases classified elsewhere with behavioral disturbance), F01.50 (vascular dementia without behavioral disturbance), F01.51 (vascular dementia with behavioral disturbance), F03.90 (unspecified dementia without behavioral disturbance), F03.91 (unspecified dementia with behavioral disturbance), and G30 (Alzheimer’s disease). We included patients with a dementia diagnosis in at least two claims during the sample selection period (*n* = 42,120) in order to increase diagnostic validity, often a concern with administrative datasets. We assigned the first date of dementia diagnosis as the index date. Of them, we further selected 36,577 patients who had at least one ophthalmologist visit within 5 years prior to the index date in order to reduce potential misclassification bias—persons with AMD identified as not having AMD due to not visiting an ophthalmologist.

Matched controls were retrieved out of the remaining NHI beneficiaries with a claim during the study period without a dementia diagnosis. Propensity score matching was used to select three controls per case. We first calculated a propensity score for each enrollee with and without a YOD diagnosis using selected demographic variables potentially associated with the diagnosis (age, sex, monthly income (NTD 0–15,840, NTD 15,841–25,000, ≥NTD 25,001; the average exchange rate in 2011 was USD 1 ≈ NTD 29), geographic location (Northern, Central, Southern and Eastern) and urbanization level of the patient’s residence (5 levels, 1 most urbanized and 5 least urbanized), and relevant medical comorbidities (hyperlipidemia, diabetes, coronary heart disease, traumatic brain injury, tobacco use disorder, alcohol dependency, obesity, and stroke). We used these selected variables in a multivariable logistic regression model to predict each selected beneficiary’s expected probability of receiving a dementia diagnosis. Because the calculated propensity score of a control patient may not exactly match that of a YOD subject, we used the method of nearest neighbor within calipers to match controls (a priori value for the calipers is ±0.01). Matching was carried out by defining the first index year of YOD patients as the year of their first YOD diagnosis. We matched three controls to a given patient with YOD based on utilization of any medical service in the index year of the YOD case. We defined each control patient’s index date as the date of their first ambulatory care claim during the matched subject’s year of diagnosis as their index date. The final study sample consisted of 36,577 YOD cases and 109,731 control patients.

### 2.3. Exposure Assessment

We identified patients with a prior diagnosis of macular degeneration using ICD-9-CM code 362.50 (macular degeneration (senile), unspecified), 362.51 (nonexudative senile macular degeneration), 362.52 (exudative senile macular degeneration), or 362.57 (Drusen (degenerative)) or ICD-10-CM code H35.30 (unspecified macular degeneration), H35.31 (nonexudative age-related macular degeneration) or H35.32 (exudative age-related macular degeneration). We defined patients as having prior macular degeneration if they had at least one claim with a diagnosis of macular degeneration during the 5 years before the index date.

### 2.4. Statistical Analysis

We carried out statistical analyses using the SAS statistical software (SAS System for Windows, vers. 9.4, SAS Institute, Cary, NC, USA). Descriptive statistics on demographics and comorbidities were summarized by counts and percentages for the YOD cases and control patients. We used chi-square tests to assess differences between cases and controls in demographic characteristics (age, sex, monthly income, geographic location and urbanization level of the patient’s residence) and medical comorbidities (hyperlipidemia, diabetes, coronary heart disease, traumatic brain injury, tobacco use disorder, alcohol dependency, obesity, and stroke). We used multivariable logistic regressions to estimate the odds ratios (ORs) and 95% confidence intervals (CI) for prior macular degeneration among patients with YOD vs. controls after accounting for age, sex, monthly income, geographic location and urbanization level of the patient’s residence, hyperlipidemia, diabetes, coronary heart disease, traumatic brain injury, tobacco use disorder, alcohol dependency, obesity, and stroke. We used two-sided *p* < 0.05 to determine statistical significance.

## 3. Results

Table 1 shows the sociodemographic characteristics and medical comorbidities among 36,577 cases and 109,731 propensity score-matched controls. Among the total 146,308 sampled patients, 64.5% were male, the majority (38.3%) resided in northern Taiwan, and only 4.4% were in southern Taiwan. Rural–urban distribution showed that the majority resided in urbanization level 2 communities (28.0%). In addition, most sample patients (45.5%) had a monthly income less than NTD 115,841.

Because we used the propensity score method to match cases to controls, we found as anticipated no statistically significant differences among most of the matching variables: age (*p* = 0.653), sex (*p* = 0.925), monthly income (*p* = 0.913), geographic location (*p* = 0.981), urbanization level (*p* = 0.970), hyperlipidemia (21.2% vs. 21.3%, *p* = 0.509), diabetes (18.2% both groups, *p* = 0.079), coronary heart disease (7.8% both groups, *p* = 0.906), traumatic brain injury (22.9%, *p* = 0.980), tobacco use disorder (16.3% vs. 16.4%, *p* = 0.683), alcohol dependency (3.8% vs. 3.6%, *p* = 0.111), obesity (18.2% both groups, *p* = 0.922), and stroke (23.6% both groups, *p* = 0.725). These results support the appropriateness of the propensity score matching process used.

The prevalence of prior macular degeneration among cases and controls is presented in Table 2, showing no significant difference (1.1% vs. 1.0%, *p* = 0.111). Univariable logistic regression analysis showed an unadjusted OR for prior macular degeneration of 1.098 among cases relative to controls (95% CI: 0.979–1.232).

Table 3 shows the results of multivariable logistic regression analysis. After adjusting for the demographic and comorbidity variables, YOD was not significantly associated with prior macular degeneration, adjusted odds ratio 1.103 (95% CI = 0.979–1.232).

## 4. Discussion

To our knowledge, this is the first nationwide population-based study exploring a possible association between macular degeneration and subsequent YOD. We found that individuals with prior macular degeneration showed no different risk for YOD (odds ratio 1.103 (95% CI = 0.979–1.232) compared to propensity score-matched controls after adjusting for sex, age group, income, geographical location, urbanization level, hyperlipidemia, diabetes, coronary heart disease, traumatic brain injury, tobacco use disorder, alcohol dependency, obesity, and stroke.

Our findings are consistent with several empirical studies that assessed the association between AMD and dementia. In the Cardiovascular Health Study Cohort recruited from a random sample of the Medicare-eligible files from four US counties, Baker et al. reported no statistically significant associations of dementia or AD with early AMD after adjustment for age, sex, ethnicity, and study center [39]. In another study using data from English national hospital episode statistics, Keenan et al. found no significant difference in the risk of dementia among patients with AMD: the rate ratio was 0.91 (95% CI, 0.79–1.04) when comparing observed vs. expected cases in the AMD cohort based on the rate observed in the reference cohort [38]. Furthermore, in a population-based, cross-sectional study of 1179 participants aged 60 to 80 years from the Singapore Malay Eye study, Ong et al. did not observe a significant association between AMD and cognitive dysfunction after adjusting for age, sex, education level, income category, and type of housing [37]. The above studies all consistently reported no association between AMD and dementia. The present study confirms these findings of a lack of association between macular degeneration and YOD.

As noted earlier, some contradicting findings are also documented. The study from Korea referenced earlier (Choi et al.) reported that AMD was associated with higher risk of subsequent AD (aHR 1.48, 95% CI 1.25–1.74) when compared to participants without AMD [33]. They suggested that patients with AMD be closely monitored for subsequent development of AD. In another case–control study using the Taiwan National Health Insurance Research Database, Tsai et al. reported that the incidence of AD or senile dementia was higher in patients with AMD than among control patients (*p* = 0.044), with an adjusted hazard ratio (HR) of 1.44 (95% CI, 1.26–1.64) after adjusting for potential confounding factors [34]. Similarly, another study from Taiwan also found that patients with AMD had a 1.23-fold increased risk of developing AD (aHR = 1.23, 95% CI = 1.04–1.46).

Unlike the prior studies, our study focused on young-onset dementia and found no association between macular degeneration and YOD. There are several possible explanations. Some studies have suggested that chronic oxidative stress and neuroinflammation, derangement of the processing and degradation of dysfunctional cellular components, and alterations of neuronal homeostasis are biological pathological mechanisms common to both macular degeneration and dementia. However, the temporal sequence remains unclear. Neuronal cell degeneration in the brain may be followed by retinal degeneration via unrelated pathways, and it remains unknown whether genetic factors, amyloid-ß deposition, oxidative stress and the associated mitochondrial and lysosomal dysfunction, or complement-related systems drive the neurodegenerative changes seen in the retina of AMD patients. Additionally, heterogeneous pathogenic mechanisms may contribute to YOD, distinct from those driving the development of senile dementia.

Our study has several strengths. Taiwan’s NHI is a highly accessible and affordable health care system for every citizen (due to negligible copayments and a widely dispersed network of physicians, almost all of them affiliated with the NHI). This minimizes the potential for diagnosis bias arising out of socioeconomic status or residential location. Further, because the NHIRD covers every medical care episode of citizens, about 23 million Taiwanese people, covering outpatient visits, emergency department visits, and inpatient admissions, NHI claims data capture the diagnoses of prior macular degeneration and YOD from all sources. Macular degeneration and YOD are highly concerning conditions that usually cause affected patients to seek medical help, which, in turn, is facilitated by affordable and accessible health care enabled by NHI. Therefore, misclassification bias is unlikely, via differential identification of macular degeneration and YOD based on socioeconomic status. Use of claims data preempts potential recall bias and social desirability bias associated with self-reported data. The case–control study design, selecting controls by propensity score matching, strengthens the study validity by minimizing selection bias and misclassification bias.

There are some study limitations. First, the diagnoses of macular degeneration or YOD were identified from the NHIRD database through the ICD-9-CM or ICD-10-CM codes captured in claims. Therefore, the study sample would have missed those who were not coded accurately. Second, the NHIRD claims lack certain critical items of data, such as physical activity, diet, cognitively challenging life activities, stress, and genetic parameters which are thought to be associated with the development of dementia. Third, the NHIRD database only includes patients who had sought care for symptoms of macular degeneration or YOD. Symptoms of early macular degeneration or YOD are often not recognized until the disease is fairly advanced. Bias from underdiagnosis and undertreatment of macular degeneration or YOD may challenge the validity of findings. Potential non-differential misclassification of YOD would have biased the results toward the null hypothesis. Fourth, patients with macular degeneration may have a great likelihood of being diagnosed with YOD because of surveillance bias, i.e., increased exposure to medical checkups and, therefore, more scrutiny of patient status and referrals (referral bias in clinical populations). As a result, the present study does not rule out the possibility that YOD detection may be enhanced among those with prior diagnosis of macular degeneration. However, our study shows no association despite possible referral bias, which adds validity to our finding. Finally, the NHIRD lacks data on the severity of macular degeneration, and it is possible that the risk of YOD may be associated with the severity of macular degeneration.

## 5. Conclusions

Based on medical claims data, our study found no evidence of an association between prior macular degeneration and YOD in Taiwan after adjusting for sex, age group, income, geographical location, urbanization level, hyperlipidemia, diabetes, coronary heart disease, traumatic brain injury, tobacco use disorder, alcohol dependency, obesity, and stroke. Despite being a population-based study, the present findings may not generalize to other countries because of differences in ethnicity and living environment. Our study suggests the need for clinical-epidemiological studies among other ethnicities and regions to extend the findings of our study.

## Figures and Tables

**Table 1 jpm-12-00291-t001:** Demographic characteristics of patients with young-onset dementia and control patients in Taiwan (*n* = 146,308).

Variable	Patients with Young-Onset Dementia (*n* = 36,577)	Controls (*n* = 109,731)	*p* Value		
	Total No.	Percent	Total No.	Percent	
Males	23,583	64.5	70,719	64.5	0.925
Age					0.653
18–44	15,218	41.6	45,507	41.5	
45–64	21,359	58.4	64,224	58.5	
Monthly income					0.913
<NTD 1–15,841	16,637	45.5	49,909	45.5	
NTD 15,841–25,000	12,226	33.4	36,578	33.3	
≥NTD 25,001	7714	21.1	23,244	21.2	
Geographic region					0.981
Northern	13,991	38.3	42,016	38.3	
Central	9696	26.5	29,080	26.5	
Eastern	11,265	30.8	33,712	30.7	
Southern	1625	4.4	4923	4.5	
Urbanization level					0.970
1 (most urbanized)	7865	21.5	23,735	21.6	
2	10,249	28.0	30,795	28.1	
3	6633	18.1	19,697	18.0	
4	5922	16.2	17,835	16.3	
5 (least urbanized)	5908	16.2	17,669	16.1	
Hyperlipidemia	738	21.2	23,393	21.3	0.509
Diabetes	6661	18.2	20,008	18.2	0.079
Coronary heart disease	2850	7.8	8529	7.8	0.906
Traumatic brain injury	8380	22.9	25,147	22.9	0.980
Stroke	8624	23.6	25,971	23.6	0.725
Alcohol abuse	1382	3.8	3990	3.6	0.211
Tobacco use disorder	5953	16.3	17,959	16.4	0.683
Obesity	6661	18.2	20,008	18.2	0.922

Note: In 2017, the average exchange rate was USD 1 ≈ New Taiwan Dollar (NTD) 30. SD, standard deviation.

**Table 2 jpm-12-00291-t002:** Prevalence, crude odds ratios (ORs), and 95% confidence intervals (CIs) for prior age-related macular degeneration among patients with young-onset dementia vs. controls.

Prior Diagnosis of Macular Degeneration	Patients with Young-Onset Dementia (*n* = 36,577)		Controls (*n* = 109,731)		*p* Value	
	*n*, Percent		*n*, Percent			
Yes	399	1.1	1091	1.0	0.111	
No	36,178	98.9	108,640	99.0		
OR (95% CI)	1.098 (0.979–1.232)		1.000			

Notes: The OR was calculated by a logistic regression.

**Table 3 jpm-12-00291-t003:** Covariate-adjusted odds of prior age-related macular degeneration (OR and 95% confidence interval, CIs) among patients with young-onset dementia vs. controls (*n* = 146,308).

Variable	Presence of Young-Onset Dementia		
	Adjusted OR	95% CI	*p* Value
Prior macular degeneration	1.103	0.982–1.238	0.535
Males	0.998	0.973–1.023	0.851
Age			
18–44	1.000		
45–64	1.053	1.026–1.081	<0.001
Monthly income			
<NTD 15,841 (reference group)	1.000		
NTD 15,841–25,000	1.011	0.984–1.039	0.439
≥NTD 25,001	0.988	0.957–1.020	0.457
Geographic region			
Northern (reference group)	1.000		
Central	0.997	0.964–1.020	0.854
Eastern	0.957	0.928–0.987	0.005
Southern	1.007	0.946–1.073	0.820
Urbanization level			
1 (reference group)	1.000		
2	0.956	0.923–0.990	0.012
3	0.970	0.932–1.009	0.130
4	0.906	0.868–0.944	<0.001
5	0.963	0.912–1.1017	0.172
Hyperlipidemia	0.988	0.957–1.020	0.459
Diabetes	0.995	0.962–1.029	0.763
Coronary heart disease	0.894	0.854–0.934	<0.001
Traumatic brain injury	0.994	0.966–1.023	0.671
Stroke	0.985	0.956–1.014	0.312
Alcohol abuse	1.046	0.982–1.114	0.161
Tobacco use disorder	1.006	0.973–1.040	0.730
Obesity	0.888	0.789–0.999	0.048

## Data Availability

Data from the National Health Insurance Research Database, now managed by the Health and Welfare Data Science Center (HWDC), can be obtained by interested researchers through a formal application process addressed to the HWDC, Department of Statistics, Ministry of Health and Welfare, Taiwan (https://dep.mohw.gov.tw/DOS/lp-2506-113.html. accessed on 2 January 2022).

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
