# Peer review of "Absence of an Association between Macular Degeneration and Young-Onset Dementia"

_jpm, 2022, doi:10.3390/jpm12020291_

Round 1
Reviewer 1 Report
In this article by Kuang et al. the authors have looked at the association between macular degeneration and young onset dementia. For this they have checked if any of the patients aged less than 65 years, were diagnosed with macular degeneration within 5 years prior to their diagnoses of young onset dementia. And the authors found that the correlation was not significantly different from that in control group.
While similar non-association has been reported before by Keenan et al., this study reports similar results in Taiwanese population for the first time.
I have following questions for the authors-
- Since the macular degenerations usually have a late onset and the authors studied younger patients, did the authors look for a reverse association i.e. was there any follow-up of the young onset dementia patients developing macular degeneration later in their life?
- Some previous studies, which the authors have also discussed in their study, have reported an association between visual impairments and neuro-degeneration in the brain because of common pathophysiology or systemic factors that affect both eye and the brain. Did the authors look at the associations between any other visual impairments, besides macular degenerations, reported in these young onset dementia patients such as glaucoma or cataracts?
Author Response
In this article by Kuang et al. the authors have looked at the association between macular degeneration and young onset dementia. For this they have checked if any of the patients aged less than 65 years, were diagnosed with macular degeneration within 5 years prior to their diagnoses of young onset dementia. And the authors found that the correlation was not significantly different from that in control group.
While similar non-association has been reported before by Keenan et al., this study reports similar results in Taiwanese population for the first time.
I have following questions for the authors-
Since the macular degenerations usually have a late onset and the authors studied younger patients, did the authors look for a reverse association i.e. was there any follow-up of the young onset dementia patients developing macular degeneration later in their life?
Response: Thanks for your suggestion. We did not investigate the relationship between young onset dementia and subsequent macular degeneration. It may take long time to develop macular degeneration following young onset dementia.
Some previous studies, which the authors have also discussed in their study, have reported an association between visual impairments and neuro-degeneration in the brain because of common pathophysiology or systemic factors that affect both eye and the brain. Did the authors look at the associations between any other visual impairments, besides macular degenerations, reported in these young onset dementia patients such as glaucoma or cataracts?
Response: The present study only focused on the association between young onset dementia and macular degeneration. We did not explore the association between young onset dementia and other visual impairments. We are sorry for that!
Reviewer 2 Report
The study proposed by Dr. Tung-Mei Tammy Kuang and colleagues is a retrospective observational analysis aimed at investigating the association between macular degeneration and young-onset dementia (YOD). The results obtained, although negative, as they do not show a correlation between the two pathologies, may be useful to the scientific community and deserve publication. However, a revision of the English language is required.
Author Response
The study proposed by Dr. Tung-Mei Tammy Kuang and colleagues is a retrospective observational analysis aimed at investigating the association between macular degeneration and young-onset dementia (YOD). The results obtained, although negative, as they do not show a correlation between the two pathologies, may be useful to the scientific community and deserve publication. However, a revision of the English language is required.
Response: We have asked the author Sudha Xirasagar, a native English speaker, to edit and revise this manuscript. Thanks a lot!
Reviewer 3 Report
The authors investigated the association between macular degeneration and young onset dementia using Taiwanese national database. The population-base database with high coverage rate and large sample size should be a reliable source. They also applied propensity score matching to adjust some confounding factors. As a result, they found no significant association between macular degeneration and young onset dementia. The result would add significant information to the controversial topic. The reviewer has only a few comments.
- Definition of each disease contain some ambiguous codes. 331.2 Senile degeneration of Brain does not necessarily accompany dementia. 362.53, 362.54 cystoid macular degeneration are not necessarily related to age-related macular degeneration. 362.55 toxic macular degeneration and 362.56 macular pucker are obviously not age-related macular degeneration.
- Although the authors stressed non-significant association, odds ratio was 1.098 and p-value was 0.11 suggesting a tendency toward association. Considering that there are some limitations to the study as discussed by the authors, it should be discussed that the result is somewhat marginal.
- About half of the discussion section is repetition of introduction section. The authors may want to focus on what was found in this study and how the result adds to the previous reports.
Author Response
The authors investigated the association between macular degeneration and young onset dementia using Taiwanese national database. The population-base database with high coverage rate and large sample size should be a reliable source. They also applied propensity score matching to adjust some confounding factors. As a result, they found no significant association between macular degeneration and young onset dementia. The result would add significant information to the controversial topic. The reviewer has only a few comments.
Definition of each disease contain some ambiguous codes. 331.2 Senile degeneration of Brain does not necessarily accompany dementia. 362.53, 362.54 cystoid macular degeneration are not necessarily related to age-related macular degeneration. 362.55 toxic macular degeneration and 362.56 macular pucker are obviously not age-related macular degeneration.
Response: Thanks for your suggestion! We have deleted ICD-9-CM code 331.2 from the selection criterion. However, we found that this group of patients with ICD-9-CM code 331.2 all have ever received a diagnosis of ICD-9-CM code 290.0.
This case-control study aimed to evaluate the association of young-onset dementia with prior macular degeneration. We did not only focus on age-related macular degeneration, so patients with ICD-9-CM codes 362.53, 362.54, 362.55 and 362.56 were included in this study.
Although the authors stressed non-significant association, odds ratio was 1.098 and p-value was 0.11 suggesting a tendency toward association. Considering that there are some limitations to the study as discussed by the authors, it should be discussed that the result is somewhat marginal.
Response: Indeed, as stated in Table 3, adjusted odds ratio was 1.103 (95% CI=0.979~1.232) and p value was 0.535. Therefore, YOD was not significantly associated with prior macular degeneration
About half of the discussion section is repetition of introduction section. The authors may want to focus on what was found in this study and how the result adds to the previous reports.
Response: Thanks for you suggestion. We have added more statements on how the result adds to the previous reports as follows: “Our findings are consistent with several empirical studies that assessed the association between AMD and dementia. In the Cardiovascular Health Study Cohort recruited from a random sample of the Medicare-eligible files from 4 US counties, Baker et al. reported no statistically significant associations of dementia or AD with early AMD after adjustment for age, sex, ethnicity, and study center [39]. In another study using the data from English national hospital episode statistics, Keenan et al. found no significant difference in the risk of dementia among patients with AMD: the rate ratio was 0.91 (95% CI, 0.79-1.04) when comparing observed vs. expected cases in the AMD cohort based on the rate observed in the reference cohort [38]. Furthermore, in a population-based, cross-sectional study of 1179 participants aged 60 to 80 years from the Singapore Malay Eye study, Ong et al. did not observe a significant association between AMD and cognitive dysfunction after adjusting for age, sex, education level, income category, and type of housing [37]. The above studies all consistently reported no association between AMD and dementia. The present study confirms these findings of a lack of association between macular degeneration and YOD.” (page 18)
“Unlike the prior studies, our study focused on young-onset dementia and found no association between macular degeneration and YOD. There are several possible explanations. Some studies have suggested that chronic oxidative stress and neuroinflammation, derangement of the processing and degradation of dysfunctional cellular components, and alterations of neuronal homeostasis are biological pathological mechanisms common to both macular degeneration and dementia. However, the temporal sequence remains unclear. Neuronal cell degeneration in the brain may be followed by retinal degeneration via unrelated pathways, and it remains unknown whether genetic factors, amyloid-ß deposition, oxidative stress and the associated mitochondrial and lysosomal dysfunction, or complement-related systems drive the neurodegenerative changes seen in the retina of AMD patients. Additionally, heterogeneous pathogenic mechanisms may contribute to YOD, distinct from those driving the development of senile dementia.”(pages 19 & 20)